# A Digital Twin Architecture to Optimize Productivity within Controlled Environment Agriculture

Jesus David Chaux, David Sanchez-Londono and Giacomo Barbieri *

Department of Mechanical Engineering, Universidad de los Andes, Bogotá 111711, Colombia; jd.chaux10@uniandes.edu.co (J.D.C.); d.sanchezl@uniandes.edu.co (D.S.-L.)
* Correspondence: g.barbieri@uniandes.edu.co

**Abstract:** To ensure food security, agricultural production systems should innovate in the direction of increasing production while reducing utilized resources. Due to the higher level of automation with respect to traditional agricultural systems, Controlled Environment Agriculture (CEA) applications generally achieve better yields and quality crops at the expenses of higher energy consumption. In this context, Digital Twin (DT) may constitute a fundamental tool to reach the optimization of the productivity, intended as the ratio between production and resource consumption. For this reason, a DT Architecture for CEA systems is introduced within this work and applied to a case study for its validation. The proposed architecture is potentially able to optimize productivity since it utilizes simulation software that enables the optimization of: (i) Climate control strategies related to the control of the crop microclimate; (ii) treatments related to crop management. Due to the importance of food security in the worldwide landscape, the authors hope that this work may impulse the investigation of strategies for improving the productivity of CEA systems.

**Keywords:** controlled environment agriculture; digital twin; productivity; architecture; optimization

## 1. Introduction

The World Bank estimates there will be a global population of over 9.6 billion people by 2050 [1]. To feed this population, agricultural production should increase by about 50% with respect to today's levels [2]. However, several aspects complicate this context. These are next illustrated using predictions to 2050. Temperature rise due to global warming will result in reduced yields. For instance, rice, maize and soybean are estimated to reduce their yields by between 3.1% and 7.4% per each degree Celsius of increased temperature [3]. The growth of the world's population will mainly occur in urban areas, while the countryside population will remain stable, i.e., from 3.4 billion in 2018 to 3.1 billion in 2050 [1,4]. Considering that food is mainly produced in the countryside, the same amount of people available today will be required to produce the necessary food. Furthermore, the lands available for agriculture will decrease by between 8% and 20% due to land degradation, urbanization, and the use of crops for biofuel production [5]. Finally, it is likely that the current threats to freshwater will determine a sub-optimal supply of water to crops [6]. The combined effect of these factors might cause a deficit of between 5% and 25% in the availability of agricultural products, threatening the food security of the planet [5]. As such, agricultural production systems need to innovate in the direction of increasing production while reducing the utilized resources.

Controlled Environment Agriculture (CEA) refers to the control of plant growth and the surrounding environment with the objective of enhancing production efficiency, optimizing plant yields, and improving product quality [7]. CEA applications—such as plant factories and greenhouses [7,8]—may become an important tool to face the aforementioned challenges. Due to their ability to control the microclimate, CEA systems are robust to climate variability and temperature rise, enabling year-round production [9]. Furthermore, they are generally associated with soilless cultures that enable the recirculation of water

and nutrients, and the utilization of inert soils [10]. In this way, the integration of CEA practices and soilless cultures provides intensive food production in limited areas with more efficient utilization of resources, including non-arable areas such as deserts and cities [11,12].

Different works have demonstrated that the increment in the level of automation determines better yield and quality of the harvested crops. Asseng et al. [13] estimated that the wheat grain cultivated through CEA vertical farmings may generate yields between $700 \pm 40$ and $1940 \pm 230$ ton/ha/yr with respect to the 3.2 ton/ha/yr obtained with open-field agriculture. Furthermore, Nicole et al. [14] detected an improvement in the lettuce food quality cultivated in plant factories with respect to open-field agriculture—in terms of color, nutrients and shelf life, amongst other things. However, the increment in the level of automation comes at the expense of higher energy consumption. Graamans et al. [15] compared greenhouses and plant factories showing that the production of 1 kg dry weight of lettuce requires an input of 247 kWhe in a plant factory, compared to 70, 111, 182 and 211 kWhe in greenhouses in, respectively, the Netherlands, the United Arab Emirates and Sweden ($\times$2).Two Sweden greenhouses were utilized in the study: One with additional artificial illumination and the other without.

From the data reported above, it can be noticed that production (both in terms of yield and quality) and energy consumption are two conflicting goals for CEA systems. To ensure food security, there is the need to reach an optimization in between these two goals by maximizing productivity: The ratio between production and resource consumption.

Digital Twin may constitute a fundamental tool to reach the optimization of productivity. Digital Twin (DT) represents the next wave in modelling, simulation, and optimization technology [16]. According to Kritzinger et al. [17] and Negri et al. [18], DT "exploits sensed data, mathematical models and real-time data elaboration to forecast and optimise the behaviour of the physical asset at each life cycle phase, in real-time". DTs have been adopted in different domains such as manufacturing, aviation, hospital management and precision medicine and safety amongst others; see [19–21].

DTs are digital models enhanced with bilateral communication between the physical and the cyber space [17]. In traditional simulation, the digital representation of an existing physical asset does not use any form of automated data exchange between the physical asset and the digital one. In a DT, the data flow between an existing physical asset and a digital one is fully integrated in both directions. In this way, the digital model is synchronized with the status of the physical asset and the results of the simulation can be directly implemented to optimize the physical asset. In the context of CEA, the DT ability to integrate the real-time status of the physical asset into simulation may be adopted to guide the decision-making in crop management and microclimate control for the optimization of productivity.

To reach the aforementioned capabilities, the physical asset must be enhanced with a DT architecture consisting of (Figure 1):

- Physical Asset: Target system to optimize through the DT architecture.
- Digital Twin: Virtual test bed synchronized with the status of the physical asset that is responsible to evaluate the different 'what-if' scenarios that may optimize the system.
- Intelligence Layer: Hosts the rules and the knowledge to choose among the alternatives tested in the DT.

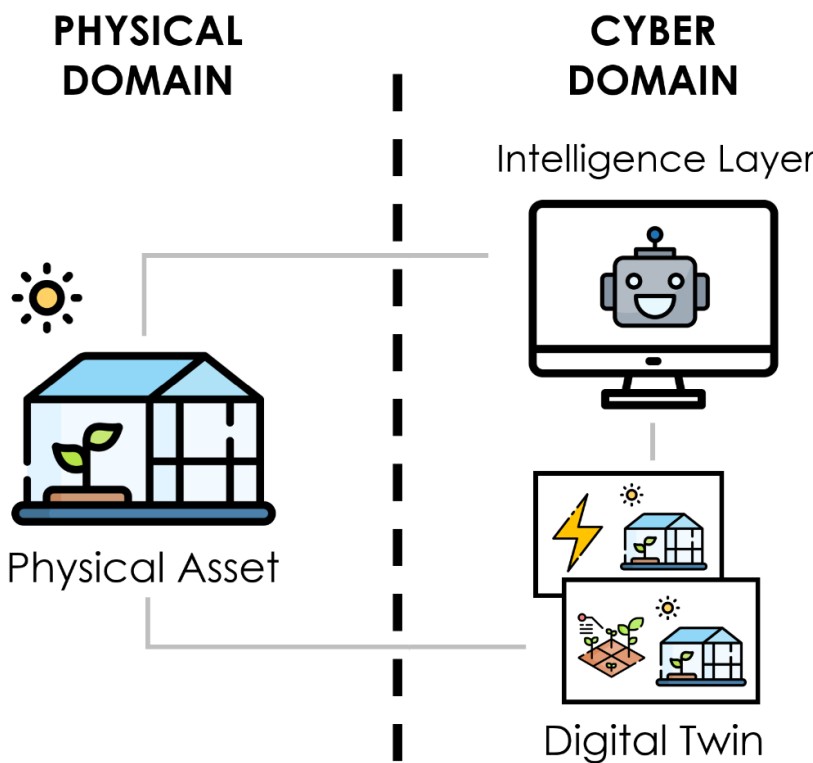

**Figure 1.** Actors involved in a DT architecture. The representation has been adopted from [22].

In the actual agricultural innovation systems, information-based management characterizes a technological phase called Farming 4.0 [23]. Few recent literature reviews can be found concerning the application of DT in agriculture. Pylianidis et al. [24] investigate the added-value of DTs for agriculture through the identification of 28 use cases, and their comparison with use cases from other disciplines. Based on their analysis, they examine the extent of the DT adoption in agriculture, shed light on the concept and the benefits it brings, and provide an application-based roadmap for a more extended adoption. Verdouw et al. [25] first review the concept of DT in agriculture by showing how DTs can advance farming practices and by developing a typology of different types of DTs. Then, they propose a conceptual framework for designing and implementing DTs. Sreedevi and Santosh [26] analyze ways in which DTs can contribute to most of the life cycle phases of hydroponic systems such as designing, operation, monitoring, optimization, and maintenance, amongst others.

In the aforementioned literature reviews, one of the potentialities of the DT for CEA applications is defined as the optimization of the productivity through the simulation and prediction of crop microclimate and growth. In their work, Rezvani et al. [27] review the basics of microclimate models for greenhouses and the results obtained from their application. Furthermore, a review of crop growth models and functional–structural plant models is also provided. Howard et al. [28] present the first advancements in the development of a DT that is intended to optimize the energy efficiency of greenhouses. Monteiro et al. [29] develop an IoT-enabled structure for vertical farming that has the objective of enabling sustainable CEA systems. Burchi et al. [30] introduce a multifunctional environment, equipped with sensors and monitoring systems that allows the acquisition of data and their processing using mathematical yield models to optimize crop management. To the best of the authors' knowledge, the presented works constitute the most relevant ones in the productivity optimization through DTs. However, it can be noticed that a DT architecture for CEA systems potentially able to optimize productivity due to the utilized simulation software is not available yet. This architecture—referred to as DT Architecture for CEA systems—constitutes the novelty presented in this work.

Given the above, the article is structured as follows: Section 2 resumes a previously introduced methodology for the design and verification of DT applications. Section 3 applies the methodology for the development of the DT architecture for CEA systems. Obtained results are discussed in Section 4, and the conclusion and future work are presented in Section 5.

## 2. Research Methodology

Barbieri et al. [31] proposed a methodology based on virtual commissioning—VC-based methodology—to retrofit manufacturing systems with DTs. The methodology consists of a stepwise approach in which the DT architecture is designed, integrated (to the retrofitted manufacturing system), and verified using a virtual environment before its implementation. In this work, part of the methodology is applied to develop the proposed DT architecture for CEA systems. Next, the research methodology utilized within this work is illustrated (Figure 2):

- Framework: The layered structure and functionalities of the DT architecture are identified without considering their implementation technologies.
- Technology: The technologies for instantiating the framework into an architecture are selected, and the actors that are interfaced within the architecture are specified.
- Digital Twin: The DT models are developed using the software and types of simulation identified in the previous phase.
- Intelligence Layer: The intelligence layer is designed starting from the defined functionalities and the selected implementation technologies. Within this phase, the interaction between the DTs and the intelligence layer is exploited with the aim to compare different optimization algorithms and to tune their parameters.
- Physical--Cyber Interface: The signals to be exchanged among the different actors within the DT architecture are identified. As depicted in Figure 1, signals are exchanged between: (i) Physical asset–intelligence layer; (ii) physical asset–digital twin; (iii) intelligence layer–digital twin. This phase also establishes the order in which signals are exchanged and which sequence of operations are implemented.
- Implementation: The architecture is implemented in the physical asset and verified.

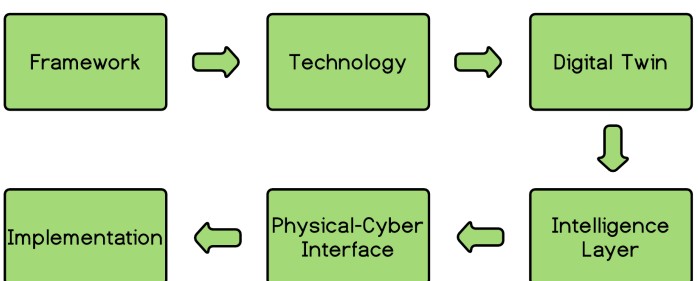

**Figure 2.** Methodology utilized for the development of the DT architecture for CEA systems. The representation has been adopted from [31].

## 3. Digital Twin Architecture

Next, the methodology illustrated in Section 2 is applied for the development of a DT architecture aimed to optimize the productivity of CEA systems. A prototype greenhouse (Figure 3) is utilized as case study for the design and verification of the DT architecture. The objective of this work is to build an architecture able to perform the bilateral communication typical of DTs using simulation models that can optimize the productivity. The utilization of the models for the identification of optimal crop treatments and climate control strategies is left as future work.

Finally, the following use case is defined for the architecture: 'The data necessary for the optimization must be available in the cloud and the user must download them in his/her local device. The optimization occurs in the local device and the optimal

crop treatment and climate control strategy are communicated to the controller for its implementation through a gateway'.

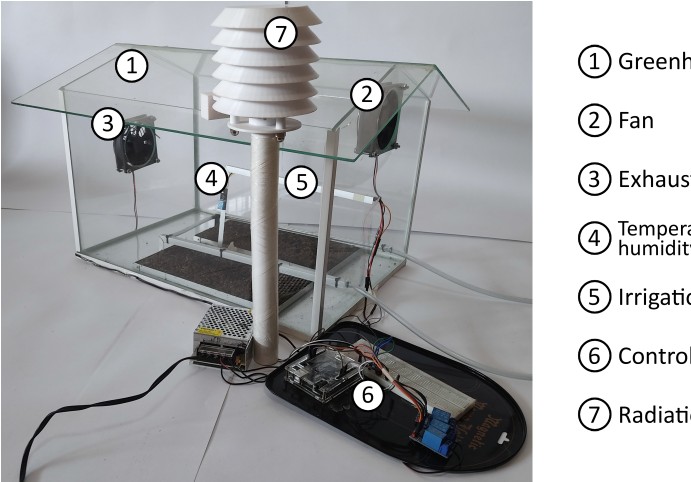

**Figure 3.** Prototype automated greenhouse utilized as case study. In the figure, the different components are indicated with numbers.

- ① Greenhouse
- ② Fan
- ③ Exhaust fan
- ④ Temperature and humidity sensor
- ⑤ Irrigation system
- ⑥ Controller
- ⑦ Radiation shield

### 3.1. Framework

In this phase, the layered structure and functionalities of the DT architecture are identified. The layered structure is depicted in Figure 4 and is illustrated next:

- Greenhouse: Physical asset to optimize; see Figure 3.
- Controller: Respectively, monitors and controls the greenhouse sensors and actuators. It also transmits the acquired sensor data to the gateway and receives the crop treatments and climate control strategies to implement.
- Gateway: Interface between the cyber and the physical domain; see Figure 1. It is responsible for transmitting sensor data to the storage layer and communicating the optimized crop treatments and climate control strategies to the controller for its implementation.
- Storage: Stores current and historical data that are utilized from the DTs for productivity optimization.
- Intelligence layer: Hosts the rules and the knowledge to choose among the different crop treatments and climate control strategies that may optimize the productivity of the greenhouse. It uses the DTs as virtual test beds to assess the evaluated alternatives.
- Digital twins: Utilizes current and historical data to assess the different crop treatments and climate control strategies received from the intelligence layer.

### 3.2. Technologies

The technologies for instantiating the framework into an architecture are selected. These are depicted in Figure 4 and illustrated next:

- Greenhouse ⟶ two DHT11 sensors to, respectively, measure indoor and outdoor temperature and relative humidity, 12 V fan and exhaust fan, and a 12 V mini submersible pump.
- Controller ⟶ Arduino Uno.
- Gateway ⟶ Raspberry Pi 4: Communicates with the storage layer through wireless communication and with Arduino through serial communication.
- Storage ⟶ phpMyAdmin: Administrator tool that manages a MySQL server for the data stored in the cloud.
- Intelligence Layer ⟶ Visual Studio: Programmed in Python, it enables the communication with the cloud through MySQL and with the gateway through the MQTT communication protocol.

- Digital Twin 1⟶ EnergyPlus (energyplus.net): Builds energy software able to predict the microclimate within the greenhouse due to the ability to simulate the behaviour of heating, cooling, ventilation, and lighting systems, amongst others. It can communicate with Python-based IDEs through the EnergyPlus API (nrel.github.io/EnergyPlus/api/python).
- Digital Twin 2⟶ DSSAT (dssat.net): Agricultural decision support system that allows the simulation of growth, development, and yield as a function of "soil–plant–atmosphere dynamics" [32]. It can communicate with Python-based IDEs through TraDSSAT (github.com/julienmalard/traDSSAT).

EnergyPlus and DSSAT were selected as simulation software for the DTs since their integration has the premises to achieve the optimization of the productivity. EnergyPlus enables the optimization of control strategies related to the control of the crop microclimate [33], whereas DSSAT enables the optimization of treatments related to the crop management [34].

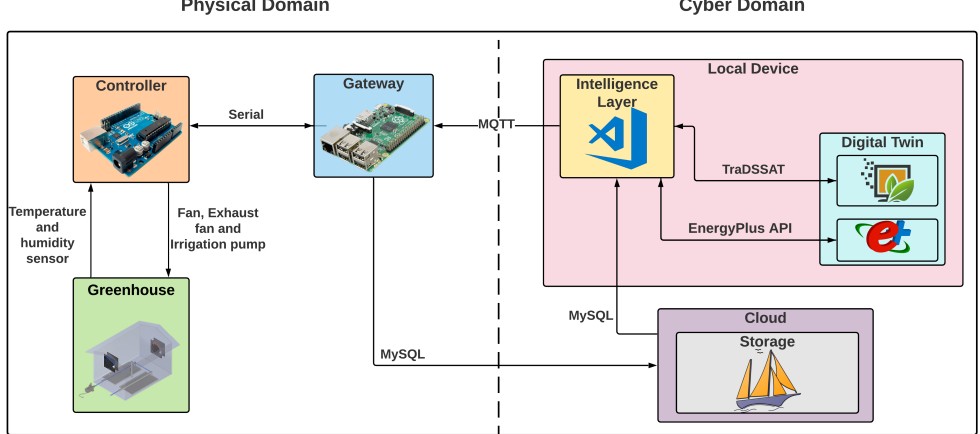

**Figure 4.** Framework and technologies utilized within the DT architecture for CEA systems.

### 3.3. Digital Twin and Intelligence Layer

After the definition of the framework and its implementation technologies, the DT models and the intelligence layer are developed. The tuning of these elements is outside the scope of the article since the objective of this work is to build a DT architecture and not to optimize the productivity of the prototype greenhouse. The demonstration of the effectiveness of the built DT architecture in optimizing the productivity is left as future work. Therefore, this section illustrates the communication between the intelligent layer and the DTs—referred to as optimization workflow—for the optimization of the productivity.

The optimization workflow is depicted in Figure 5 and consists in the following phases:

1. Generation of climate control strategies: The intelligence layer receives from the cloud: (i) Microclimate historical data (internal temperature and relative humidity); (ii) environmental historical data (external temperature and relative humidity); (iii) previous climate control strategies and crop treatments. Starting from these data, different alternatives of climate control strategies (CCSs) are generated. In this work, a CCS is defined as a control sequence of the greenhouse actuators to achieve a desired crop microclimate. Then, a prediction of the future environmental conditions is performed since EnergyPlus needs this information to assess the different CCSs.
2. Assessment of climate control strategies: The historical microclimate data, and the historical and predicted environmental data are transmitted to EnergyPlus. Then, all the generated CCSs are input to the software, and the predicted energy consumption and microclimate are computed for each CCS using the historical and predicted climate data.

3. Generation of crop treatments: The intelligence layer receives the predicted energy consumption and microclimate relative to each CCS. Then, different alternatives of crop treatments (TRTs) are generated, e.g., event to trigger the irrigation, volume delivered for irrigation, etc.

4. Assessment of crop treatments: The historical and predicted microclimate data are transmitted to DSSAT. Then, all the generated TRTs are input to the software, and the predicted production and resource consumption (e.g., water, nutrients, etc.) are computed for each TRT using the historical and predicted microclimate data.

5. Overall optimization: The intelligence layer receives the predicted production and resource consumption relative to each TRT. The productivity is computed for each pair of CCS and TRT—where the productivity is defined as the ratio between the production, and the sum of the energy and resource consumption. The best pair of climate control strategy ($CCS^*$) and crop treatment ($TRT^*$) is computed and transmitted to the gateway.

Given the interdependence between EnergyPlus and DSSAT, an overall optimization must be implemented. Starting from the domain knowledge, optimization algorithms and/or heuristics should be studied as future work to identify optimal solutions in acceptable computation time.

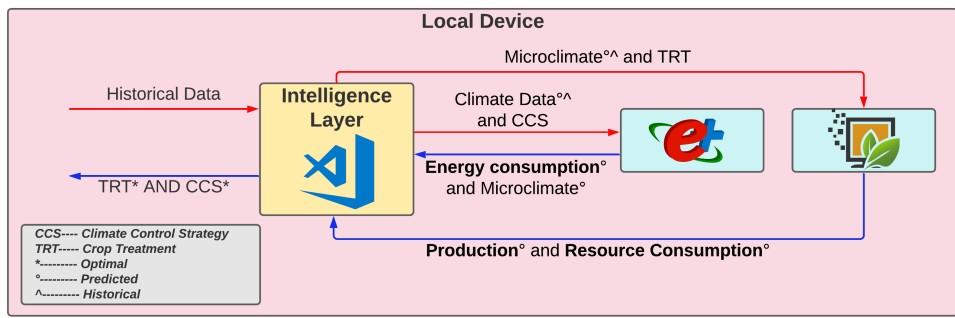

**Figure 5.** Optimization workflow: Communication between the intelligent layer and the DTs for the optimization of the productivity. Production and energy and resource consumption are indicated in bold since they are utilized for the calculation of the productivity. In the figure, climate data indicate the set of microclimate and environmental data.

### 3.4. Physical–Cyber Interface

After the design of the optimization workflow, the signals to be exchanged among the different actors within the DT architecture are identified. In this phase, the order in which signals are exchanged is also established. This information is depicted in Figure 6 through a sequence diagram. Sensor data are assessed from the controller and continuously uploaded to the cloud through the gateway. When an optimization is performed, historical data are sent to the local device and the optimization workflow illustrated in Section 3.3 is implemented. Once the optimal $CCS^*$ and $TRT^*$ have been identified, these are transmitted to the gateway. Finally, the gateway sends them to: (i) Cloud: To trace the implemented climate control strategies and crop treatments; (ii) Controller: To implement the optimal climate control strategy and crop treatment.

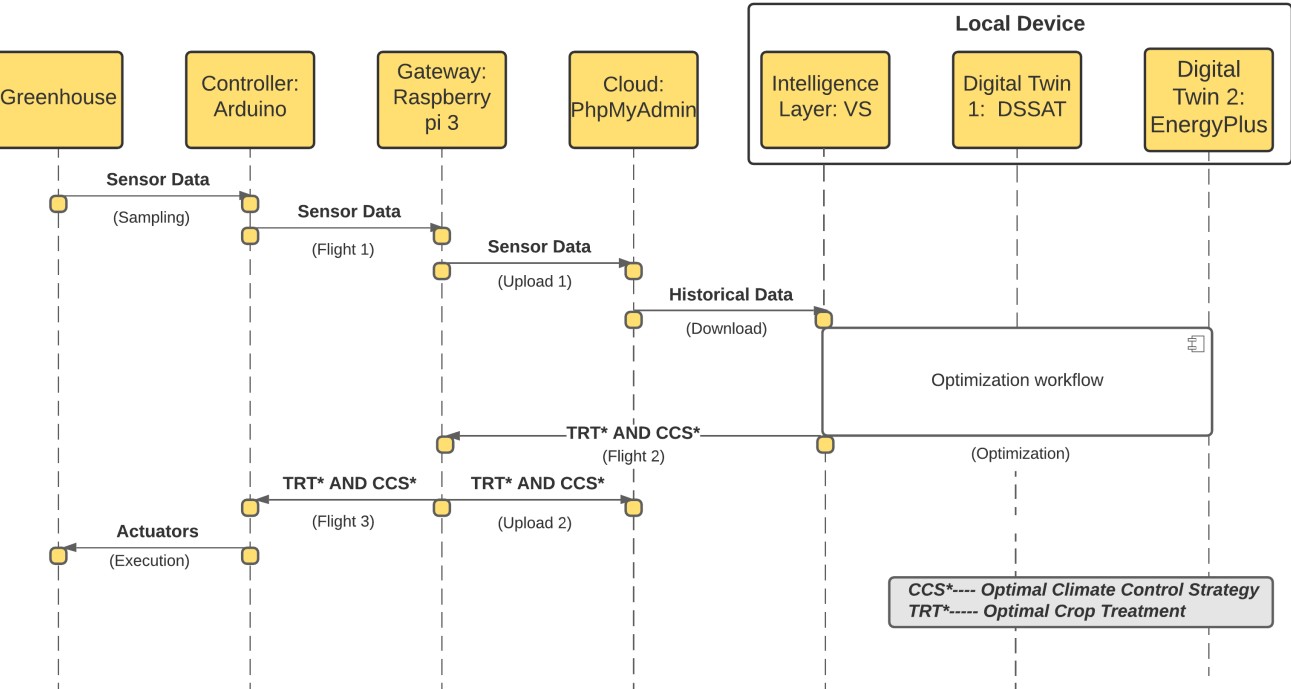

**Figure 6.** Sequence diagram showing the communications between the actors of the DT architecture. The name of each communication is indicated in parentheses.

### 3.5. Implementation

The presented DT architecture is applied to the prototype greenhouse illustrated in Figure 3. Then, latency tests are performed to verify the implementation of the communications depicted in Figure 6.

## 4. Results and Discussion

Latency tests were performed to verify the implementation of the DT architecture. In this article, a latency test refers to the assessment of the time taken for a message to travel in between two actors. It is worth noting that the minimization of the latency was not within the scope of this work and does not constitute a priority for CEA production systems since these are not hard real-time ones. Latency was assessed as a mean to certify the achievement of the communications indicated in Figure 6.

Figure 7 illustrates the latency taken within each communication. To emulate a feasible optimization scenario, a single data point was transmitted within each communication. The only exception was constituted by the 'download' communication in which one million data points were downloaded from the cloud. The communication sequence illustrated in Figure 6 was repeated 300 times to evaluate the latency variability through the random error formulation [35].

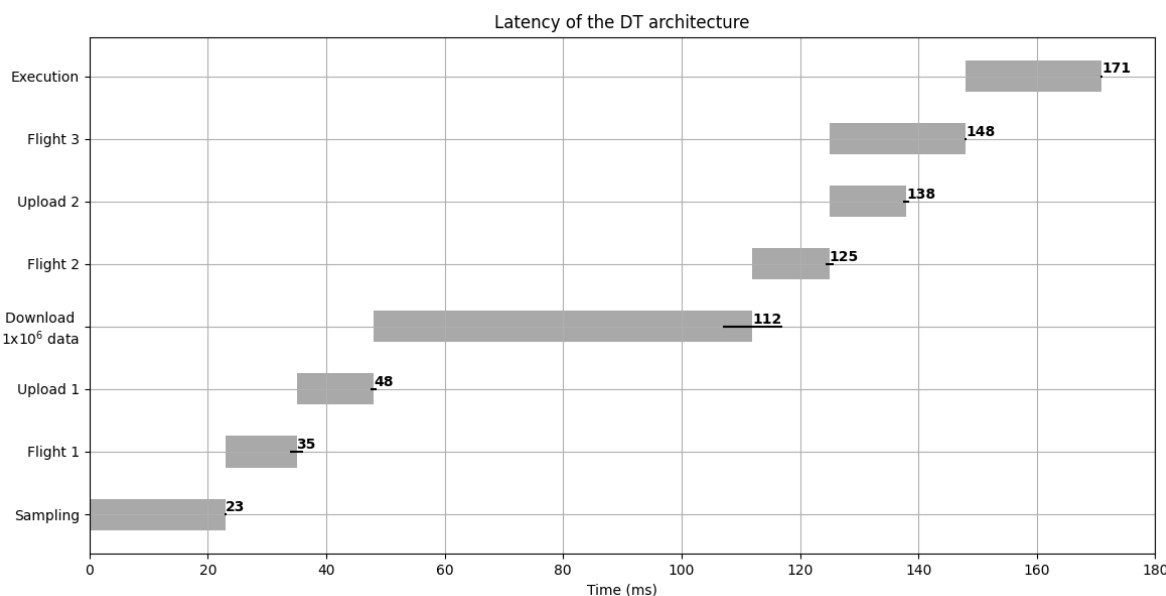

**Figure 7.** Timing diagram of the latency taken within each communication. Error bars quantify the uncertainty of each measurement. The nomenclature shown in Figure 6 is utilized for defining the different communications.

The latency data demonstrate that it was possible to achieve the bidirectional communication typical of DT architectures [17], i.e., from the physical domain to the cyber one and vice versa (Figure 1). Considering the selected simulators, a DT architecture potentially able to optimize the productivity was built.

From an industrial perspective, the proposed DT architecture constitutes a scalable architecture that may be applied to industrial systems by replacing the Arduino controller with the utilized PLCs (Programmable Logic Controllers). From an educational perspective, the illustrated low-cost prototype may be replicated from educational institutions for the generation of automation and agricultural hands-on laboratories, and for the investigation of approaches for the optimization of the productivity in CEA production systems.

Even if a functional DT architecture was built, several works are still missing before certifying its ability to optimize the productivity. In particular, a case study should be implemented and productivity should be optimized through the developed architecture. Different challenges will occur as the aforementioned definition of heuristics for the identification of close-to-optimal climate control strategies and crop treatments in acceptable computation time.

## 5. Conclusions and Future Work

To ensure food security, agricultural production systems should innovate in the direction of increasing production while reducing the utilized resources. Controlled environment agriculture and digital twins may represent fundamental tools to reach the optimization of productivity, thus contributing to the planet's food security.

With this in mind, the objective of this research work was the development of a DT architecture potentially able to optimize productivity in the context of CEA applications. The objective was reached by designing an architecture that utilizes (as DTs) simulation software that enables the optimization of: (i) Climate control strategies related to the control of the crop microclimate; (ii) treatments related to the crop management. The architecture was applied to a prototype greenhouse for its validation. Finally, communication latency was assessed as a means to test the achievement of the communications defined within the DT architecture.

This work contributes to the research on DT in CEA systems by proposing an architecture potentially able to optimize productivity. The methodological approach and the identified tools can be utilized by companies for retrofitting their CEA systems with the

DT functionality, and from universities for the generation of automation and agricultural hands-on laboratories, and for the investigation of approaches for the optimization of the productivity in CEA systems.

However, several works are still missing before the ability of the proposed architecture to optimize the productivity can be certified. In line with this goal, some future works are identified:

- Optimization workflow: The optimization workflow identified within this work involves a sequence of two simulation software. Starting from the domain knowledge, optimization algorithms and/or heuristics must be studied to identify optimal solutions in acceptable computation time.
- Case study: After the definition of heuristics, a case study must be implemented and productivity must be optimized to certify the ability of the proposed DT architecture to optimize productivity.
- Architecture improvement: Some improvements should be investigated as the movement of the intelligence layer and the DTs to the cloud, and the simplification of the physical domain with the implementation of smart sensors and actuators that would make the controller and gateway unnecessary.

**Author Contributions:** Conceptualization, J.D.C. and D.S.-L.; methodology, J.D.C. and G.B.; software, J.D.C.; validation, J.D.C. and D.S.-L.; formal analysis, J.D.C. and D.S.-L.; investigation, J.D.C., D.S.-L. and G.B.; resources, G.B.; data curation, J.D.C. and D.S.-L.; writing—original draft preparation, J.D.C., D.S.-L. and G.B.; writing—review and editing, G.B..; visualization, J.D.C. and D.S.-L.; supervision, G.B.; project administration, G.B.; funding acquisition, G.B. All authors have read and agreed to the published version of the manuscript.

**Funding:** This research received no external funding.

**Institutional Review Board Statement:** Not applicable.

**Informed Consent Statement:** Not applicable.

**Data Availability Statement:** Not applicable.

**Conflicts of Interest:** The authors declare no conflict of interest.

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
