# Peer review of "A Digital Twin Architecture to Optimize Productivity within Controlled Environment Agriculture"

_applsci, doi:10.3390/app11198875_

Round 1

Reviewer 1 Report

Only minor comments:

1) It would be good to have a diagram with indicative KPIs and their related thresholds

2) BAck up the "Np-hardness" of the problem with a reference

Author Response

The authors would like to thank the reviewer for reviewing the manuscript, and raising constructive questions, which led to the improvement of the article. The following responses have been prepared to address the reviewer’s comments in a point-by-point fashion. In addition, a new version of the manuscript has been uploaded. We have tried to address all the points and believe that the revised version can meet the publication requirements. All amendments made have been highlighted in blue color in the revised paper.

Review comments:
1) It would be good to have a diagram with indicative KPIs and their related thresholds

We thank the reviewer for the constructive recommendation. After a deep analysis, we decided to not include a table with relevant KPIs and thresholds. The reason is that it is difficult to standardize such a table generating a useful content. In fact, KPIs and thresholds concerning environmental variables, resource consumptions and target production rates and quality depends on the considered cultivated crops.  

2) Back up the "Np-hardness" of the problem with a reference

Thank you for this observation. We used the NP-hard word without making sure that the problem was effectively NP-hard. In order to not insert a wrong information, we decided to review this part by defining it as a complex problem and not as an NP-hard one. In the manuscript, now we have: 

"Given the interdependence between EnergyPlus and DSSAT, an overall optimization must be implemented. Starting from the domain knowledge, optimization algorithms and/or heuristics should be studied as future work to identify optimal solutions in acceptable computation time."

Author Response

The authors would like to thank the reviewer for reviewing the manuscript, and raising constructive questions, which led to the improvement of the article. The following responses have been prepared to address the reviewer’s comments in a point-by-point fashion. In addition, a new version of the manuscript has been uploaded. We have tried to address all the points and believe that the revised version can meet the publication requirements. All amendments made have been highlighted in blue color in the revised paper.

Review comments:

1) The abstract can be improved by making its structure sounder: study problem and background first, then aim, methods, inputs and main results and findings. The latter should be emphasized.

We thank the reviewer for the constructive recommendation. However, we think that the abstract includes all the points mentioned in the comments and already present the result with the proper emphasis. Next, we demonstrate this:

Study problem and background: Increase production while reducing resources
To ensure food security, agricultural production systems are called to innovate in the direction of increasing production while reducing the utilized resources. Due to the higher level of automation with respect to traditional agricultural systems, Controlled Environment Agriculture (CEA) applications generally achieve better yields and quality crops at the expenses of higher energy consumption. 

Aim, methods: Digital Twin
In this context, Digital Twin (DT) may constitute a fundamental tool to reach the optimization of the productivity, intended as the ratio between production and resource consumption. For this reason, a DT Architecture for CEA systems is introduced within this work and applied to a case study for its validation. 

Results and findings: Architecture potentially able to optimize productivity
The proposed architecture is potentially able to optimize productivity since utilizes simulation software that enables the optimization of: (i) climate control strategies related to the control of the crop microclimate; (ii) treatments related to the crop management.

2) Add the words and references: In the actual agricultural innovation systems, information-based management is characterizing a technological phase called Farming 4.0 [Vecchio et al., 2020].  

We thank the reviewer for the constructive recommendation. We included the suggested review in the manuscript

3) Add the words and references: and safety amongst others [19; Agnusdei et al., 2021a; 2021b].

We thank the reviewer for the constructive recommendation. We included one of the two suggested references. Placing two references just for safety would not be fail for the other application domains. 

4) The Conclusions section is too short and must be rewritten in order to emphasize the contribution of the study and discuss its specific findings.

We thank the reviewer for the constructive recommendation. We add the following paragraph to enphatize the contribution of the study and discuss its specific findings:

"This work contributes to the research on DT in CEA systems by proposing an architecture potentially able to optimize productivity. The methodological approach and the identified tools can be utilized from companies for retrofitting their CEA systems with the DT functionality, and from universities for the generation of automation and agricultural hands-on laboratories, and for the
investigation of approaches for the optimization of the productivity in CEA systems.

However, several works are still missing before certifying the ability of the proposed architecture to optimize the productivity. In line with this goal, some future works are identified:"

5) An extensive editing of English language and style by a native speaker is required.

We thank the reviewer for the constructive recommendation. We reviewed and improved the English of the paper. For the sake of simplicity, we have not noted all the language and syntax correction in the new version of the article.

Reviewer 3 Report

The authors studied digital twins to collect a real-time data to predict the plant needs based on climatic conditions in the greenhouse. 

This manuscript reports a very preliminary study.

In title of the manuscript "optimize productivity" was cited. I was expecting a greenhouse test with real plants and a system which control environmental conditions e.g. nutrients, solar radiation, and interaction between soil and air humidity. If possible, the authors could include these information because are important agronomic production variables.

Plants responses (productivity, nutrient accumulation, enzymes activities) could be evaluated and would be interesting to supply the cyber domain. Can the system understand/process the environment data and provide the water, solar radiation, nutrients as the plant need? 

In my opinion, the most concerning point of the manuscript is that only theoretical information was provided, which is important for future studies. The proposed system should be compared with the already established automatization system that is used in commercial farms.

Specific comments:
Line 56: Where do they used with or without additional illumination? Sweden, UAE or Netherlands? 
Or for lettuce in a plant factory they used artificial illumination and greenhouses they did not used artificial complementation?
Please, clarify the sentence.

Author Response

The authors would like to thank the reviewer for reviewing the manuscript, and raising constructive questions, which led to the improvement of the article. The following responses have been prepared to address the reviewer’s comments in a point-by-point fashion. In addition, a new version of the manuscript has been uploaded. We have tried to address all the points and believe that the revised version can meet the publication requirements. All amendments made have been highlighted in blue color in the revised paper.

Review comments

1) In title of the manuscript "optimize productivity" was cited. I was expecting a greenhouse test with real plants and a system which control environmental conditions e.g. nutrients, solar radiation, and interaction between soil and air humidity. If possible, the authors could include these information because are important agronomic production variables.

We thank the reviewer for this observation. Within the description of the software utilized as digital twins, we think that this information is already available:

EnergyPlus: building energy software able to predict the microclimate within the greenhouse due to the ability to simulate the behaviour of heating, cooling, ventilation and lighting systems amongst others
DSSAT: Agricultural Decision Support System that allows the simulation of growth, development and yield as a function of ”soil-plant-atmosphere dynamics”

2) Plants responses (productivity, nutrient accumulation, enzymes activities) could be evaluated and would be interesting to supply the cyber domain. Can the system understand/process the environment data and provide the water, solar radiation, nutrients as the plant need? 

Thanks for this question. We think that the answer is already included within the manuscript. The simulators utilized as Digital Twin are potentially able to identify climate control strategies (values of the actuators to reach target values for internal microclimate variables; e.g. humidity, temperature, solar radiation, etc.) and treatments; i.e. water, nutrients, etc. This information is explained within section 3.3.

3) The proposed system should be compared with the already established automatization system that is used in commercial farms.

Thanks for the suggestion. However, we think that the manuscript in its actual form already include this information. The majority of commercial systems do not include digital twins and the ones that has this, do not use simulators able to optimize both production and resource consumption.

We think  that this sentence placed at the end of the introduction remarks this concept:
"it can be noticed that a DT architecture for CEA systems potentially able to optimize productivity due to the utilized simulation software is not available yet."

5) Line 56: Where do they used with or without additional illumination? Sweden, UAE or Netherlands? Or for lettuce in a plant factory they used artificial illumination and greenhouses they did not used artificial complementation? Please, clarify the sentence. 

Thanks for the question. The with or withouth additional illumination is referred to the Sweden greenhouses. We changed the sentences and now it should be clear.

Two Sweden greenhouses were utilized in the study: one with additional artificial illumination and the other without.

Round 2

Reviewer 3 Report

The manuscript has been improved according to the suggestions.

I still have the opinion that this is a preliminary study, despite knowing the importance of the topic and the study per se.

I suggest a reformulation in the title to inform the readers that further studies have not been performed. 

Suggestion: A Digital Twin Architecture to Simulate Productivity Optimization within Controlled Environment Agriculture 

Or other title that make it clear that study in greenhouse with "real plants" has not been performed yet.